# Leveraging National Nursing Home Huddles for Rapid COVID-19 Response

**DOI:** 10.3390/geriatrics6020062

**Published:** 2021-06-17

**Authors:** Rebecca Brandes, Elias Miranda, Alice Bonner, Joelle Baehrend, Terry Fulmer, Jennifer Lenoci-Edwards

**Affiliations:** 1Institute for Healthcare Improvement, Boston, MA 02109, USA; emiranda@ihi.org (E.M.); jbaehrend@ihi.org (J.B.); jlenoci-edwards@IHI.org (J.L.-E.); 2The John A. Hartford Foundation, New York, NY 10022, USA; terry.fulmer@johnahartford.org

**Keywords:** COVID-19, nursing facilities, huddles, communication, community, learning

## Abstract

A disproportionate number of older adult residents of U.S. nursing homes have died during the COVID-19 pandemic. The novelty of the virus spurred frequently changing guidance as nursing facilities navigated response efforts. In May 2020, the 6-month COVID-19 Rapid Response Network for Nursing Homes (RRN) was launched to leverage the concept of huddles across U.S. nursing homes to reduce COVID-19-related morbidity, mortality, and transmission by identifying best practices to rapidly implement, fostering connections between nursing homes, and refocusing the national narrative on optimism for nursing home care response efforts. Daily 20-min huddles transitioned to twice weekly in the program’s final two months. A total of 93 huddles featured 103 speakers with 1960 participants engaging in both live huddles and asynchronous learning. 90.33% of participants said they learned at least two new ideas by participating and 89.17% strongly agreed or agreed that participating improved their ability to lead change in their organization. Qualitative data echoed gratitude for a centralized source of information and best practices and the sense of positivity and community the RRN provided. Leveraging nursing home huddles at the national, regional, local, system, or facility level may serve as a guidepost for future pandemics or work where guidance is new or quickly evolving.

## 1. Introduction

A disproportionate number of older adult residents of U.S. nursing homes have died and are dying during the COVID-19 pandemic. As of 25 November 2020, more than 100,000 long-term care facility residents and staff have died of COVID-19, accounting for 40% of all COVID-19 deaths [1]. Nursing homes often feature communal living spaces and house older adults with co-morbidities, making them hotspots for increased COVID-19 risk [2]. The novelty, sudden onset, and deadliness of the virus contributed to an environment of urgency across nursing homes in the United States. Nursing home leaders, responsible for the safety of their residents and staff, worked to stand up and quickly adapt their COVID-19 response efforts in the face of rapidly changing guidance and often with a lack of necessary resources, such as adequate personal protective equipment (PPE).

Huddles are a well-known communication tool in health care that enable unit-level staff to exchange information collaboratively and efficiently. The huddles format of brief, frequent touchpoints to resolve patient issues fosters open communication and can improve patient safety and outcomes [3,4]. Advocate Health Care, for example, implemented daily, 15-min safety huddles in 2013, which increased safety event reporting by 40% and dramatically decreased safety events by 2014 [5].

In May 2020, with funding from The John A. Hartford Foundation (JAHF), the Institute for Healthcare Improvement (IHI) launched the COVID-19 Rapid Response Network for nursing homes (RRN) to reduce COVID-19 related morbidity and mortality, adapting and scaling the huddles concept as a daily virtual touchpoint for nursing home leaders and staff across the U.S. The RRN was intended to reduce transmission by connecting nursing home leaders and staff across the country to share their learning and experience with one another, sharing timely data and policy updates, and harvesting and providing information about best practices for nursing homes to rapidly implement. A related goal of the initiative was to change the national narrative around nursing home care from one focused on despair to a story of optimism by highlighting the important and difficult work undertaken in nursing homes across the country to keep residents and staff safe despite unprecedented challenges. The free 6-month network featured 20-min virtual huddles on focused COVID-19 clinical or operational topics driven by the needs identified by participants. The huddles were held daily for the first 4 months and twice weekly for the final two months of the RRN.

While the program was launched with a stated aim of reducing COVID-19 related morbidity, mortality, and transmission, several factors limited our ability to track those outcomes and measure progress toward that aim. These factors included a decision to not collect data from participating nursing homes so they could focus their efforts on responding to the immediate crisis, a recognition that the Centers for Medicare & Medicaid Services was already requesting real-time data on top of their normal data collection, and the number of confounding variables at play during a quickly evolving pandemic. Therefore, the aim of this paper is to describe the huddles model used in the COVID-19 RRN, the experiential results and impact from participating nursing homes, and future possibilities for the scaled-up huddles format.

## 2. Materials and Methods

### 2.1. Marketing and Recruitment

IHI leveraged outreach strategies to drive participation in the RRN, which was free to join. Strategies included promoting the program to existing IHI database contacts with nursing home-related activities or interest, leveraging existing communication channels and newsletters, and creating paid social media retargeting campaigns based on relevant job titles, industry types, and locations, as well as individuals with similar behaviors or profiles to those currently in the database. We also approached experts in the field and relevant associations, membership groups, regional networks, and large systems with requests to join or spread the word. Colleagues at The John A. Hartford Foundation leveraged their networks and provided support and expertise to spread word about the RRN program and what it offered.

Participants enrolled in the RRN were automatically added to an email listserv and provided access to a materials landing page with connection information, huddle recordings, and resources. Engagement was measured by the total number of enrollments, number of people who joined each huddle, and visits to the materials landing page.

### 2.2. Huddles Design

Huddles were 20 min in length to balance having enough time to communicate useful information and minimizing the burden on participants’ workload with residents. To facilitate nursing home staff engagement, IHI emailed eight nursing homes for insight on the optimal time of day for huddles. Noon Eastern Time was identified as the preferred time due to its alignment with the lunch hour and morning shift change at many nursing facilities.

IHI established a standard agenda for huddles to reduce variation, increase efficiencies for the planning team, and create a sense of familiarity for participants. Table 1 outlines each agenda item, its duration, and purpose.

The RRN huddles used the Cisco Webex Event Center video conferencing platform to allow for sharing slides, participant chat, video display of speakers, as well as polling and recording capabilities.

The most important design principle was that the huddles reflect and support participants’ needs and evolving environment. While relying heavily on standardization and reducing variation, we launched the huddles with an open mind for redesign and provided regular opportunities for participants to provide feedback about content and format.

### 2.3. Internal Resourcing

A team of five IHI staff hosted the huddles. The facilitator role rotated between four staff and included introducing each agenda item and their associated speakers. The role of the Webex host, which included technology set up and troubleshooting, was rotated between two staff members. Role delineation, rotation, and use of standardized scripts alleviated the burden of IHI staff needing daily planning calls and ensured a seamless experience for participants through staff vacations and unexpected sick time.

Other IHI staff roles included marketing and communications, research, speaker management, participant communications, measurement and evaluation, and content curation.

### 2.4. Speakers and Topics

Huddles featured expert speakers for the data minute, policy update, and best practice segments. In an effort to standardize, the data minute was regularly led by experts from the Centers for Disease Control and Prevention (CDC) and the policy update was led by experts from The Society for Post-Acute and Long-Term Care Medicine (AMDA). Best practice speakers were sourced through research, existing relationships, word of mouth, and open invitations to participants.

The driver diagram in Figure 1 depicts our theory of the drivers that will contribute to reducing COVID-19-related mortality in nursing homes. It was created after a research scan conducted in May 2020 and refined throughout the 6-month RRN. The driver diagram was created to inform the topics and best practices to be featured during huddles, so the program covered a comprehensive and effective array of information. Topics were also sourced through participant surveys and themes from participant questions during huddles.

While identification of speakers and topics for huddles required advance planning, we avoided confirming them too far in advance to allow the RRN to truly be shaped by participant needs and the evolving environment.

### 2.5. Communications and Continuous Improvement

Communications were intentionally designed to reach nursing home participants who were busy providing care to residents. Automatic email reminders were sent the morning of each huddle to keep the huddles top of mind and Webex connection information easily accessible. Each Friday, a weekly wrap-up email outlined the topics covered and pointed to resources on the materials page for those who missed the huddles or who wanted to review or share with colleagues. In addition to recordings, slides, and resources shared during the huddles, the materials page also included a one-page summary of key themes from each week.

As stewards of continuous improvement, IHI staff designed regular methods of obtaining feedback from participants. Each huddle ended with a quick poll asking participants to rate the huddle’s usefulness and provided an open space for feedback. As the RRN progressed, the poll evolved, changing to three times per week and replacing the usefulness prompt with a call for questions relating to the week’s topic. A more comprehensive survey was sent to participants once a month, with the option to respond in real time during huddles using Webex’s polling function, or following huddles using a Microsoft Forms survey. Monthly questions included an RRN usefulness rating, inquiry on impact of the huddles, feedback on the various huddle segments and topics, and open space to share desired topics and types of speakers.

## 3. Results

The RRN huddles were offered daily on each weekday from 4 May until 21 August, 2020. After Week 16 and a two-week redesign pause, the huddles occurred twice weekly, on Tuesdays and Thursdays, from 8 September to 29 October 2020. The redesign was in response to a small drop-off in live huddle attendance and participant response to a polling question about optimal frequency and preferred days of the week for future huddles. Across the 6 months, we held 93 20-min huddles with 1960 total participants and 105 speakers featured. The following sections dive further into our results.

### 3.1. Participation and Engagement

#### 3.1.1. Participation

At the time of the last huddle on 30 October 2020, there were 1960 participants enrolled in the RRN. Figure 2 shows that enrollment increased sharply in the first two months, with more than 1000 people enrolled within the first two weeks. Later, enrollments plateaued, a trend that matches the intensity of the marketing and recruitment efforts as well as the novelty-driven urgency of COVID-19.

The nearly 2000 RRN participants comprised individuals with roles such as nursing home administrators, executive directors, nurse managers/practitioners, and quality improvement managers from organization types including skilled nursing facilities, academic health institutions, and rehabilitation centers. 1691 participants provided their location upon enrolling in the RRN, which showed representation across all 50 states in the U.S. Figure 3 depicts the spread of participants across the country, with high participation from Texas, Massachusetts, New York, and California. While not pictured, Alaska and Hawaii had 14 and 8 participants, respectively. 36 participants reportedly joined from outside of the U.S., mostly from Canada and the United Kingdom.

#### 3.1.2. Engagement

Attendance for the RRN huddles was strongest in the initial weeks of the program, with the first week having the highest average live attendance at 240.8 participants (*n* = 5) for each huddle. Figure 4 depicts the average live huddles attendance by week for the 6-month program (May to October 2020). The first few weeks were followed by a decline in live attendance before plateauing at 130 attendees. For Weeks 1–16, during which daily huddles occurred, each huddle averaged 150.81 participants (*n* = 77). Attendance began to decline in the weeks before the program redesign at Week 17, and maintained steady after twice-weekly huddles were implemented. Over the course of the complete 24 weeks, huddles averaged 130.25 participants (*n* = 93). Based on Webex attendance reports and chat logs, we believe the huddles developed a consistent attendee base with many “regulars.”

While the RRN huddles occurred daily through August, the most highly attended day of the week was Monday, averaging 180.27 participants (*n* = 15), followed by Tuesdays, which averaged 164.75 participants (*n* = 16). Figure 5 shows the average attendance by day of the week for the first 16 weeks of huddles, indicating declining attendance throughout the week.

Participants who did not join the live huddles were able to access recordings, resources, and email communications for asynchronous participation. Figure 6 shows visits to the materials page that houses the recordings and resources; visits peaked in Week 2 when the page was launched. The materials page received an average of 354 weekly page visits (*n* = 24) throughout the 6 months. There was a small spike in materials page visits during Week 10, which was devoted to quality assurance and performance improvement (QAPI) and the 4Ms Framework of an Age-Friendly Health System [6]. Materials page visits decreased in Week 17 after reducing huddles from daily to twice weekly, with less frequent updates to the page (from daily to twice weekly).

### 3.2. Speakers and Topics

Throughout the first 15 weeks of huddles, a unique topic was selected each week on which best practice speakers focused their presentations. Topics included COVID-specific guidance on PPE, COVID-19 testing, infection prevention, reopening guidance, and topics related to the 4Ms Framework of Age-Friendly Care (what matters, medication, mentation, and mobility) [6]. Topics also included key issues that nursing homes expressed as being top of mind, including workforce and staffing, caring for residents with dementia, the emotional well-being of staff and residents, and the role of certified nursing assistants (CNAs). Week 2 received the highest live engagement when the huddles agenda focused on testing. Other high points of engagement included Week 3, focused on the emotional well-being of staff and residents; Week 6, focused on infection prevention and control; and Week 12, which highlighted workforce and staffing. After the redesign, huddles were held twice weekly and featured a combination of returning speakers and assorted topics suggested by the nursing homes. The disproportionate impact that COVID-19 has had on communities of color was a frequent topic throughout the duration of the huddles. Many data updates and a week of the program’s best practice speakers in July was devoted to equity. Table 2 provides the complete list of topics addressed during the 6-month RRN program.

The best practice speakers for each huddle came from a variety of backgrounds and included academics, nursing home administrators, leaders of CNA associations, and infection preventionists. See Appendix A for a complete list of huddle speakers.

### 3.3. Experiential Results

Table 3 provides a selection of qualitative experiential results from end-of-month surveys completed by participants. Participants enjoyed the brevity of the 20-min huddles and the structure and the flexibility it allowed them, with some suggesting huddles extend to 30 min. Participants expressed satisfaction with huddles and indicated that they shared the Friday weekly wrap-up emails and materials page resources with their colleagues. Participants also expressed enjoyment with the positivity of huddles and the community the huddles built that facilitated connections, shared learning, and a feeling of not being alone at work. Figure 7 illustrates the most valuable agenda segments selected by participants in month-end surveys. Participants indicated that the best practice segment was the most valuable component of RRN huddles, followed by the policy update and the data minute.

### 3.4. Impact Results

In the month-end surveys submitted by participants, the RRN received positive feedback from participants. On average, in each of the monthly surveys (*n* = 6), 90.17% of participants surveyed said huddles had been extremely useful or useful. 89.17% of participants agreed or strongly agreed that participating would improve their ability to lead change in their organization. 90.33% of participants said they had learned at least two new ideas by participating in the huddles in the previous month.

In a final evaluation of the RRN, we asked participants to share something they had learned in the huddles that they brought back to their organization or incorporated into their work. Many attendees replicated the huddle system in their settings and found that the huddle structure contributed to improved communication among staff. Table 4 outlines a selection of ideas that participants learned in huddles and brought back to their organizations, and Table 5 shares some quotes relating to impact results from end-of-month participant surveys. Overall, participants increased information sharing across their facilities, adopted huddles or augmented their current ones, and made improvements to their COVID response as a result of the RRN.

## 4. Discussion

The scaled nursing home huddles model developed and implemented in the RRN may serve as a guidepost for future public health events or large-scale work where guidance is new or quickly evolving. In the event of a future pandemic or similar public health crisis, the model provides a centralized source of information, access to experts, and a community that fosters hope and shares learning and best practices. The model allows for the ability to rapidly address and adjust to the evolving needs of participants, and the brevity of the format provides flexibility for participants to be able to easily attend or catch up asynchronously. The data minute and policy update were critical to keeping participants abreast of the latest COVID-19 trends and recommendations; the best practice segment provided participants with concrete guidance on how to implement new practice in a rapidly evolving environment; and sharing ‘one good thing’ helped bring optimism to a difficult time.

While some aspects of the format of huddles, such as duration, frequency, and dates, may not have been ideal for every participant, selecting the most preferred and including asynchronous participation options made the huddles successful. Asynchronous options and the short 20-min agenda were also key to mitigating the load on staff time. Public health crises like COVID-19 can stretch the resources of a facility, and participation in a daily huddle can easily stretch staff even thinner. For example, we heard that it was difficult for CNAs, incredibly busy staff members, to make it to the huddles reliably. For this reason, it was vital that the huddles remain valuable and worthwhile. One way that facilities made this work with their available resources, which we learned in survey feedback, was that some participants sent representatives to the huddle who digested and communicated the information to others in their organization. Overall, future iterations of the scaled huddles model should expect and plan for general attendance to decline over time due to the sustainability of joining at a daily frequency and as the need for information in the environment decreases. Providing various methods for obtaining and incorporating feedback in an ongoing way, and being open to design changes throughout the process, ensures that huddles are valuable to participants and meet the community’s needs. Regular summaries, such as the RRN’s weekly wrap-up emails, can help participants glean key takeaways and prioritize asynchronous resources. Finally, offering a variety of expert speakers (with varied perspectives, experiences, roles, and backgrounds) and identifying relevant topics that address participants’ needs can help encourage long-term participation.

We hypothesize, based on participant engagement via chat and feedback in polls and surveys, that participants experienced a high level of psychological safety and a meaningful sense of community in huddle. Frequent expressions of thanks and encouragement for those working to improve care for residents fostered an atmosphere of mutual admiration and respect across participants, guest speakers, and the IHI team. The positive nature of the huddles kept attendees engaged and providing several feedback touchpoints helped the IHI team ensure that the content was relevant and meaningful to participants.

The RRN huddle model need not be scaled at a national level; it could be considered regionally, locally, or across facilities within a system. Individual facilities might also consider implementing daily huddles and some aspects of the RRN huddle design, as many of our participants did. Experts from the field or individuals within a community, system, or facility might use the huddles model to share updates on the latest data and policies. Various caregivers could be highlighted to share best practices and case examples. Interested caregiver champions could take on the facilitator role on a rotating basis—a great opportunity for developing leadership within the organization. We encourage anyone implementing huddles, at any scale, to test it for a month, adapt the design to their context, create inclusive leadership roles, and think through how to measure and evaluate the huddles on an ongoing basis.

The driver diagram served as an important frame as we determined topics, speakers, and best practices to feature in each huddle. The driver diagram supported the program in covering a comprehensive set of strategies that all aligned with the ultimate goal of reducing COVID-19 related morbidity, mortality, and transmission. At the same time, feedback from huddle participants and expert presenters informed our further development of the driver diagram. For example, a number of presentations on how to provide psycho-emotional support to staff during COVID-19 provided important change ideas related to the ‘effective leadership and management’ driver. Professional associations we remained in contact with beyond the program, such as the American Medical Director’s Association (AMDA), the Gerontological Society of America (GSA) and LeadingAge, continue to come back to these concepts during biweekly meetings.

Two critical elements for huddles are an intentional focus on equity and quality improvement. A full week of daily huddles were offered for each topic (equity and QAPI) in addition to weaving in relevant content throughout the program. Addressing systemic inequities within nursing facilities and building the quality improvement capability of caregivers can foster impactful change. Making equity an explicit topic of discussion helps ensure that nursing homes continue to find ways to address inequities in their settings. Grounding the huddles in improvement science was key to fostering an environment to spur new ideas and promote change within participants’ settings. In the survey data, 90.33% of RRN participants stated that they learned two or more ideas from the huddles, and 89.17% shared that attending the huddles improved their ability to lead change in their organization. Improving the experiences and outcomes of nursing facility residents was the driving theme behind every huddle, and RRN participants embraced that spirit to spark positive change in their settings. A future area of study for the huddles might include engaging participants in improvement projects in their settings that can be tracked throughout the duration of the huddles.

A limitation of the RRN was the inability to track the reduction of COVID-19 related morbidity and mortality as a result of implementing the huddle model. It is likely that participating nursing homes improved care, and therefore resident outcomes, to some extent using the knowledge, resources, and increased communication fostered by the RRN; however there is a lack of data to provide more specific evidence of impact. Future implementations of scaled huddles need to consider focusing on establishing resident outcome measures and tracking data and improvements over time.

## 5. Conclusions

The huddle model for rapid learning utilized by the RRN can be leveraged nationally, regionally, locally, or across facilities within a system to provide up to date guidance and foster a sense of positivity and community during future public health events where guidance is new or quickly evolving. As older adults are particularly vulnerable to infectious diseases such as COVID-19, having a centralized source of information that is easily accessible to nursing home staff is vital to providing the best care. The model not only provides staff with guidance and improvement ideas but can also facilitate increased information sharing and improved communication in nursing facilitates. Designing the huddles to be responsive to the needs of its participants can strengthen engagement and the spirit of collaboration throughout the program.

## Figures and Tables

**Figure 1 geriatrics-06-00062-f001:**
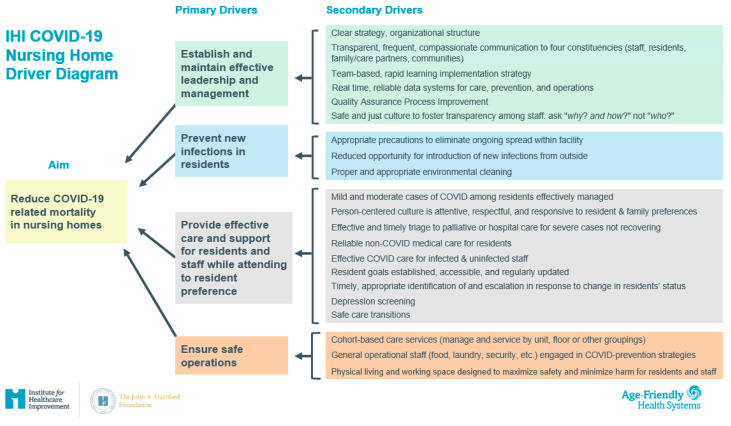
Driver diagram on reducing COVID-19 mortality in nursing homes.

**Figure 2 geriatrics-06-00062-f002:**
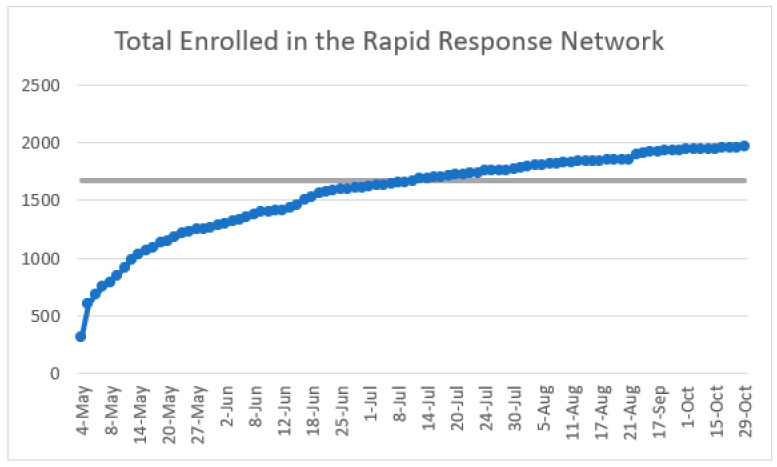
Total participant enrollment in the RRN (May–October 2020).

**Figure 3 geriatrics-06-00062-f003:**
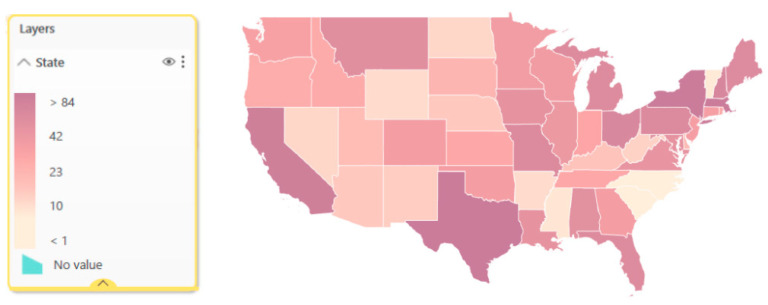
Participation in the RRN by U.S. State (excluding Hawaii and Alaska).

**Figure 4 geriatrics-06-00062-f004:**
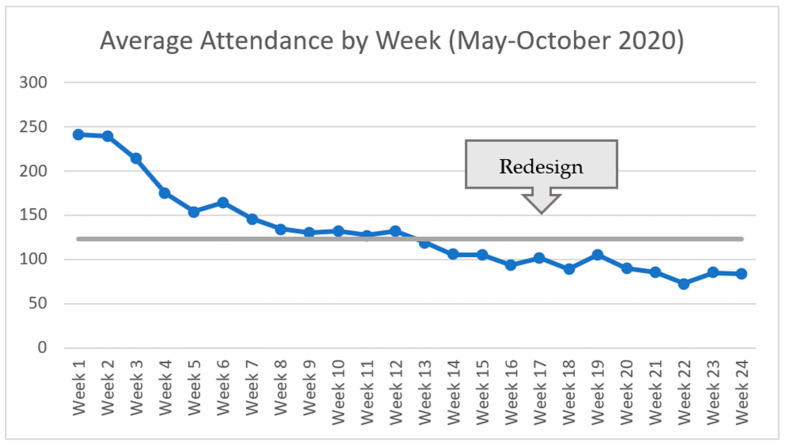
Average live RRN huddle attendance by week (May–October 2020).

**Figure 5 geriatrics-06-00062-f005:**
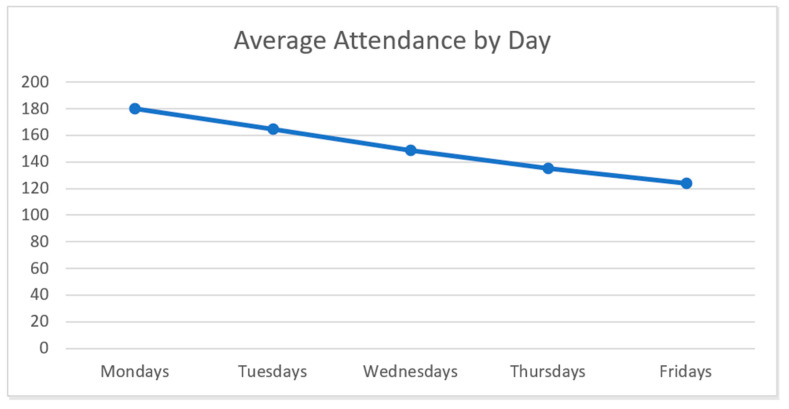
Average RRN huddle attendance by day (May–August 2020).

**Figure 6 geriatrics-06-00062-f006:**
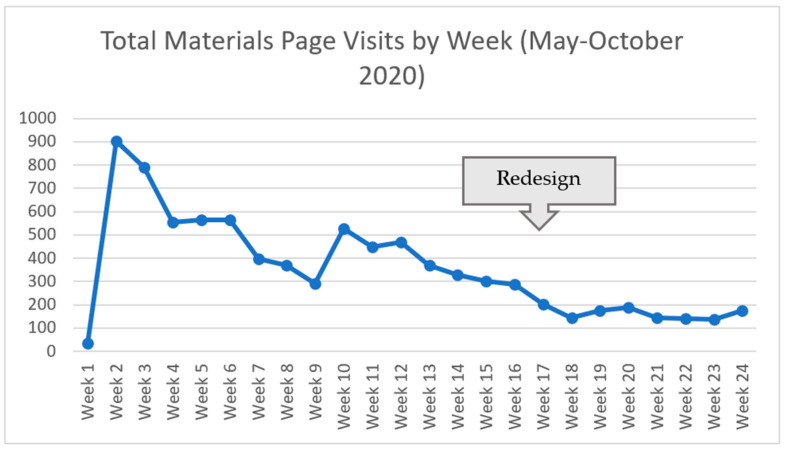
Total RRN materials page visits by week (May–October 2020).

**Figure 7 geriatrics-06-00062-f007:**
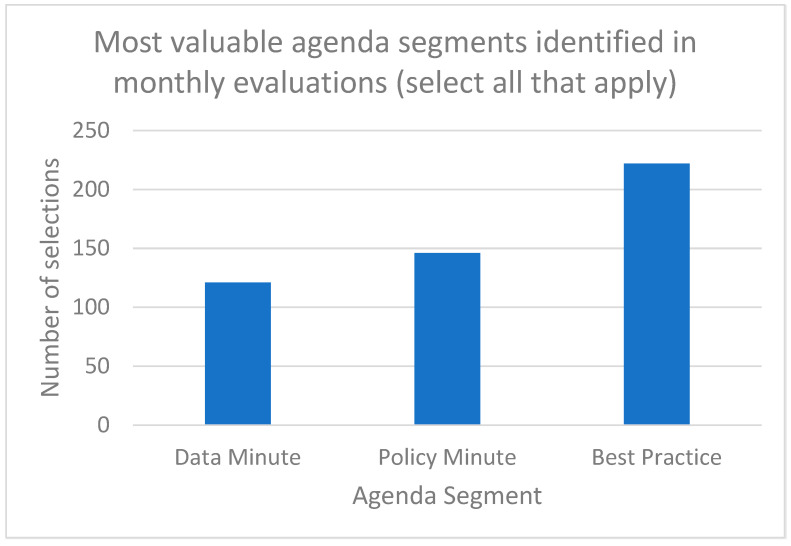
Most valuable huddle agenda segments identified in monthly participant surveys (*n* = 6).

**Table 1 geriatrics-06-00062-t001:** Standard agenda for RRN huddles.

Time	Agenda Item	Purpose
1 min	Welcome	Introduce the call and orient to the agenda
3 min	One Good Thing	Create space for participants to share ‘one good thing’ that has happened to them recently to foster optimism
1 min	Data Minute	Communicate the latest, relevant COVID-19 data
2 min	Policy Update	Update participants on relevant COVID-19 policies
8 min	Best/Better Practice	Highlight bright spots and practices to avoid through examples from the field
2 min	Closing	Share closing reminders and collect feedback
3 min	Buffer time	Allow time for transitions and flexibility to ensure a prompt stop at the 20-min mark

**Table 2 geriatrics-06-00062-t002:** RRN huddle topics by week (May–October 2020).

Dates	Topic
Week 1: 4–8 May 2020	Personal Protective Equipment (PPE)
Week 2: 11–15 May 2020	COVID-19 Testing
Week 3: 18–22 May 2020	Emotional Well-Being of Staff and Residents *
Week 4: 25–29 May 2020	Mobility
Week 5: 1–5 June 2020	What Matters
Week 6: 8–12 June 2020	Infection Prevention and Control *
Week 7: 15–19 June 2020	Continuum of Care
Week 8: 22–26 June 2020	The Role of CNAs
Week 9: 29 Jun–3 July 2020	Advance Care Planning
Week 10: 6–10 July 2020	QAPI and the 4Ms of Age-Friendly Care
Week 11: 13–17 July 2020	Reopening Guidance
Week 12: 20–24 July 2020	Workforce and Staffing *
Week 13: 27–31 July 2020	Equity *
Week 14: 3–7 August 2020	Caring for Residents with Dementia
Week 15: 10–14 August 2020	Age-Friendly Health Systems *
Weeks 16–26: 17 August–30 October 2020	“Best of” and Other Emerging Topics

* = Indicates a favorite topic in the given month based on end-of-month participant survey.

**Table 3 geriatrics-06-00062-t003:** Selection of RRN huddle qualitative experiential results from monthly surveys.

Theme	Quotes from Participant Survey Responses	Month
Format	“I think the huddles are wonderful…20 min is the perfect time.”	5
	“The format is excellent and the calls have really helped me as a beginner.”	5
	“These are very valuable...and seem to go by fast—wonder if you could extend to 30 min?”	1
	“The programming changes have been really good, you are obviously listening to feedback. I completely love the one-page summary at the end of the week.”	1
Positivity	“The positivity and the upbeat nature of these huddles has been inspiring.”	6
	“The huddles are the highlight of my day.”	3
	“I love ‘one good thing.’ It is sometimes challenging to think of something good. These are tough times, so thank you for the challenge.”	3
	“Keep up the great work—this has been very good for my mental health.”	1
Community	“Connecting with others is the most helpful part of these huddles.”	6
	“It’s reassuring to know we are not alone in our work.”	1
	“So appreciate that you are doing this. I feel supported by this network, in the same boat as everyone else. Truly appreciate it.”	1

**Table 4 geriatrics-06-00062-t004:** Selection of ideas learned in huddles that participants brought to their organizations.

Ideas Learned in RRN Huddles
Calculate PPE burn rate
Use of MyStory tool to promote resident-centered care
Develop cohorting strategies to reduce transmission
Implement huddles agenda for communication and problem solving
Create name badges with staff photos to reduce resident confusion, isolation
Use clear masks for deaf or hard-of-hearing residents
Initiate peer support groups to alleviate caregiver stress
Plan parades and other distanced events for resident socialization

**Table 5 geriatrics-06-00062-t005:** Selection of qualitative impact results from monthly participant surveys.

Theme	Quotes from Participant Survey Responses	Month
Increased Information Sharing	“I share at least one huddle each week with our health care management team to implement.”	6
	“I spent time with a nurse yesterday sharing many mobility tips.”	1
	“I have recommended the recorded huddles and resources to facilities we serve and our partners and clinicians, since the information and excellent resources are useful and have across-the-board applications.”	4
	“I have been sharing (the weekly wrap-up email) with my organization each week.”	5
Adoption of/Augmenting Huddles	“The IHI huddle set the pattern for communication with our teams. The simple agenda is easy to follow and appropriate for the remote environment with our nurse practitioners and changed how we approach our remote meetings. We use the data minute, the policy update, and then enlist staff members to share a patient story to anchor our shared commitment to long-term care patients.”	6
	“I incorporated the use of one good thing into our team COVID huddles. Implementing a positive energy into the meetings made positive difference and helped develop interfacility relationships.”	6
	“We have increased the number of huddles that we are having from shift change to adding huddles at 11 AM, 2 PM, and 5 AM so that leadership can touch base with staff to see how their shift went and the struggles they faced.”	6
Improvements to COVID Response	“Appreciate up-to-date changes and tools to work with, considering so much is changing.”	1
	“After this week’s huddles, I reviewed dialysis patients for proper virus prevention; tightened up infection control procedures; started planning more staff team building efforts.”	1
	“I learned about the importance of thoroughly assessing, testing, and proactive planning in advance of transitioning our residents to a different level of care. The huddles have also helped me keep in mind that we must always look at the ‘big picture’ with regard to the care and safety of each resident during this pandemic, as well as to encompass the resident’s and family’s point of view and preferences in an effort to ease their anxiety and fears.”	6
	“The information provided related to improvement science and the (IHI) Psychology of Change Framework [7] has been inspirational in my quest to improve and to assist me in creating an enthusiasm to improve in my workplace. The tools provided allow for easy (or easier) understanding for all, even those without ‘improvement’ backgrounds or experiences.”	6
	“Based on a huddle that took place during CNA week, my facility elected to bring on a CNA supervisor. Beyond management, the CNA supervisor was put in place to implement solutions to mitigate burnout and mental fatigue among CNAs during the COVID pandemic.”	6

## Data Availability

Quantitative data from this study are available in a publicly accessible repository here. Qualitative data are available on request from the corresponding author. Qualitative data are not publicly available due to having being compiled from multiple surveys and sources.

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
