# Peer review of "Leveraging National Nursing Home Huddles for Rapid COVID-19 Response"

_geriatrics, 2021, doi:10.3390/geriatrics6020062_

Round 1
Reviewer 1 Report
This well-written paper reports on a rapid response network (RRN) for nursing homes that utilized twice weekly huddles to provide information, networking, and illuminate the positive work of participating nursing homes in response to the Covid-19 pandemic. The RRN operated virtually with an asynchronous option. The experiential results, impact upon nursing homes, and future possibilities are described in this paper.
The introduction provides excellent context and rationale for the project. Citations are appropriate .
The methods associated with recruiting, implementing, and evaluating the RRN are sound and described thoroughly. The continuous improvement approach is particularly laudable. This section provides an excellent blueprint for replication.
Results are clearly presented and included data about participation, content of the huddles, and the qualitative feedback of the participants, that addressed the impact of the project and lessons learned.
The discussion section flows from the results and synthesizes findings with recommendations for future iterations. MINOR RECOMMENDATION: The authors should also consider/discuss if and how the RRN approach may address other crises faced by nursing homes.
Author Response
- Point 1: This well-written paper reports on a rapid response network (RRN) for nursing homes that utilized twice weekly huddles to provide information, networking, and illuminate the positive work of participating nursing homes in response to the Covid-19 pandemic. The RRN operated virtually with an asynchronous option. The experiential results, impact upon nursing homes, and future possibilities are described in this paper. The introduction provides excellent context and rationale for the project. Citations are appropriate. The methods associated with recruiting, implementing, and evaluating the RRN are sound and described thoroughly. The continuous improvement approach is particularly laudable. This section provides an excellent blueprint for replication. Results are clearly presented and included data about participation, content of the huddles, and the qualitative feedback of the participants, that addressed the impact of the project and lessons learned. The discussion section flows from the results and synthesizes findings with recommendations for future iterations.
- Response 1: Thank you for your review and for this feedback.
- Point 2: MINOR RECOMMENDATION: The authors should also consider/discuss if and how the RRN approach may address other crises faced by nursing homes.
- Response 2: Some of this was mentioned in the Discussion (for instance, the types of events or crises that may warrant the use of the model and ways of adapting the model). We made a change to the Discussion section to add clarity and strengthen the statement.
Reviewer 2 Report
Dear authors, your paper contains exciting information. However, contrary to the aim, the paper does not describe the impact of the huddles as presented in Figure 1. The paper presents participants' opinions in the huddles (impressions) and not impact - reducing COVID19 related morbidity and mortality, or nurses' capacity. In fact, it would be beneficial for readers to present your study results concerning your theory of the drivers - structure the results according to primary and secondary drivers. Also, were there any disadvantages of participating in huddles during the COVID-19 pandemic? What were the organization and safety events in these nursing homes before, during, and after the 6-month huddles sessions? How huddles impacted the nursing tasks and organisation? etc.
Thank you.
Author Response
- Point 1: Your paper contains exciting information. However, contrary to the aim, the paper does not describe the impact of the huddles as presented in Figure 1. The paper presents participants' opinions in the huddles (impressions) and not impact - reducing COVID19 related morbidity and mortality, or nurses' capacity. In fact, it would be beneficial for readers to present your study results concerning your theory of the drivers - structure the results according to primary and secondary drivers. What were the organization and safety events in these nursing homes before, during, and after the 6-month huddles sessions? How huddles impacted the nursing tasks and organisation? etc.
- Response 1: Thank you for your review and for this feedback. While the ultimate reason for launching the Rapid Response Network was to reduce COVID-19-related morbidity, mortality, and transmission, we were not set up to track those outcomes nor safety events and nursing tasks throughout the program. We make mention of this limitation at the end of our Discussion; however, we thank you for the opportunity to strengthen the section and discuss the topic earlier in the paper. We added some language to the end of the Background to make sure this is clear. The driver diagram was created to help guide and shape a comprehensive selection of best practices and speakers for the huddles. We added in some language to clarify its purpose and use in Materials and Methods section 2.4 and in the Discussion.
- Point 2: Also, were there any disadvantages of participating in huddles during the COVID-19 pandemic?
- Response 2: We appreciate this prompt. We included a note about the disadvantages, staff time attending huddles during a crisis where resources are already stretched, to the Discussion section for your review.
Round 2
Reviewer 2 Report
Dear authors, my congratulations for an inspiring paper.